**www.cambridge.org/ext**

## Perspective

de-extinction; ecosystem function; ecosystem services; ethics and policy; megafauna

**Corresponding author:**
Paul R. Jepson;
Email: prjepson@gmail.com

# De-extinction beyond species: Restoring ecosystem functionality through large herbivore rewilding

Paul R Jepson 

PJ Consulting, Musselburgh, East Lothian, United Kingdom of Great Britain and Northern Ireland

## Abstract

This perspective positions rewilding as a novel approach to ecosystem restoration, emphasising the restoration of natural processes to create self-willed ecosystems. Central to European rewilding is the de-domestication of cattle and horses to act as functional analogues of the extinct aurochs and wild horses. This de-extinction pathway shifts the focus from the loss of species to the loss of their ecological roles caused by human actions commencing millennia ago. The focus on restoring functional effects provides a strong policy rationale for large herbivore de-domestication, aligning with nature-based solutions to address environmental challenges. This alignment requires a pragmatic approach that prioritises the restoration of ecosystem functions over genetic purity and offers flexibility and scalability in rewilding efforts. I argue that creating a new category of 'ecosystem engineer' livestock is more effective than seeking wild status for these animals. As they are released into recovering ecosystems, de-domesticated large herbivores are recreating their ecological roles, 'life-spheres' and interactions. These processes open new avenues in both extinction discourse and ecological theory and encourage us to explore how de-extinct species can drive the recovery of European ecosystems.

## Impact statement

In the context of European rewilding, the purpose of mega-herbivore de-extinction is to create a new breed of 'ecosystem engineer' livestock with phenotypes that replicate the functional effects of their extinct ancestors. These animals can serve as key allies in the accelerated recovery of ecosystems, acting as assets for nature-based solutions to complex environmental and social challenges. Rewilding not only provides a clear policy rationale for de-extinction but also opens new frontiers for de-extinction research.

## Introduction

Extinction is generally viewed as the endpoint in the process of population decline. In conservation policy discourse, this process is often attributed to human causality, indifference, and irresponsibility but rarely implies conscious purpose or intent. In contrast, de-extinction signifies a process with a human-defined purpose. This includes the moral desire to right past wrongs, to restore species with cultural significance and/or that contribute to ecosystem function, and to push the boundaries of science, particularly genetic engineering and biotechnology. In the context of rewilding, the primary purpose and motivation is to restore taxa that played key roles in the functioning and evolution of ecosystems and whose return is expected to accelerate the recovery of ecosystem integrity.

Rewilding is recognised as a progressive approach within the field of restoration ecology. Unlike traditional ecological restoration that aims to restore degraded ecosystems to a 'natural' or cultural reference condition, rewilding eschews a defined end-point and is more open-ended (Gann et al., 2019; Mutillod et al., 2021). The primary aim of rewilding is to restore natural processes, interactions, and non-human autonomy to support the recovery of creating self-willed, self-sustaining, and resilient ecosystems while reducing human pressure and control (Prior & Ward, 2016; Svenning, 2020; Jepson, 2022). It is a future-oriented approach, aligned with contemporary understandings that ecosystems are not static but constantly in flux and shaped by interactions between species, processes, and their environment. In the European context, rewilding is closely aligned with the discourse of nature-based solutions: it aims to restore ecosystems as assets and infrastructure for climate adaptation, nature-based enterprises, rural regeneration, biodiversity recovery, and more broadly, the transition to a post-fossil carbon civilisation.

Central to the European rewilding approach is the restoration of megafauna guilds (Svenning et al., 2016). This is a response to the recognition that in many biomes ecosystems and megafauna co-evolved, However, human hunting during the late Pleistocene/Early Holocene caused megafauna extinctions and the extinction of the ecological interactions created by megafauna (and in particular mega-herbivore guilds). This has caused a 'down-grading' of ecosystem complexity and integrity (Lorimer et al., 2015). In Europe, the majority of the late Pleistocene megafauna survive, but cattle and horse taxa only as domesticated breeds.

## Extending the concept of extinction to ecosystem functional effects

Large herbivore rewilding extends the concept of extinction from the loss of individual taxa to the loss of functional animals within ecological systems. This rewilding approach offers an applied basis for advances in functional ecology and trait-based analysis, potentially contributing to a re-theorisation of conservation science. A significant milestone in this respect was a 2014 international conference in Oxford, UK, which explored how megafauna extinctions have altered ecosystem structure and function, as well as the patterns and consequences of megafauna decline. This led to special features in *PNAS* and *Ecography*. Synthesising insights from these studies, Malhi et al. (2016) concluded that the Late Pleistocene and subsequent megafaunal extinctions had profound effects on: (i) the physical structure and dynamics of ecosystems; (ii) vegetation composition, (iii) trophic cascades, and (iv) ecosystem biogeochemistry.

Trophic rewilding emphasises the restoration of megafauna guilds to restore ecosystem complexity (Svenning 2020). The focus on large-bodied animals stems from the critical role that body size plays at higher trophic levels. Large herbivores have *vegetation structure effects* through grazing, browsing, and trampling that alter grass-woody vegetation dynamics and create micro-habitat diversity leading to primary and secondary consumer abundance. They cause *terrain and hydrological effects* through wallowing and rooting soil that disturbs and aerates soil and creates ephemeral water bodies creating microhabitats and promoting soil, seed generation and others Additionally, they have *dispersal and biochemical effects* from defecation, birthing, dying and roaming that facilitate processes of seed, microbe and nutrient dispersal facilitating plant growth and providing and food sources for scavengers and decomposers. Such functional effects shape and drive ecosystem processes, function, structure and complexity: their decline and extinction have reduced ecosystem integrity and, in many regions, led to ecosystem phase-shifts.

The magnitude of the effects caused by large herbivores' is a function of the morphological and behavioural traits of each species interacting with specific environmental contexts. Domestication has Thanks 'downgraded' these traits and their interactions with ecological systems, such that the IUCN classifies both the Auroch (*Bos primigenius*) and wild horse (*Equus ferus*) as extinct. However, whilst domestication of horses began approximately 5,500 years ago on the Eurasian steppes and domesticated herds subsequently spread across Europe, many of their ecosystem functional effects have survived. This is because the social behaviour of horses afforded free-ranging husbandry models, allowing wild and domestic herds, along with their genes and behaviours, to intermingle over millennia. Traditions of keeping horses in semi-feral states survive to this day in parts of Eastern and Southern Europe, as well as Iceland (Lovász et al., 2021; Linnartz et al., 2023). These semi-feral horse populations retain many of the phenotype traits and functional effects of their wild ancestors. From a rewilding perspective, their 'de-extinction' involves restoring foraging, roaming, defensive, and intra-specific behaviours that arise from interactions with another megafauna in more structurally diverse ecosystems. These intra-specific interactions, which include predation, modulate the spatial magnitude of the functional effect types described in the preceding paragraph.

In contrast, the domestication of Europe's wild cattle, the Auroch (*B. primigenius*), involved a deliberate separation of wild and domestic populations, driven by the desire to breed cattle to maximise the yield of meat, milk, leather products and draft power services. This domestication process significantly changed phenotype leading to a loss of ecosystem effects and the development of highly modified grassland habitats to maximise production within modern livestock systems. The Auroch de-extinction process has focused on creating a new breed—the Tauros—that expresses the morphological, physiological, and behavioural traits of its wild ancestor. This process follows a de-domestication pathway involving: (1) a back-breeding programme that selects animals from primitive cattle herds retaining Auroch-like traits, guided by advances in genomic science that can simulate morphology using the four available Auroch genomes (Park et al. 2015); and (2) the introduction of Tauros herds into recovering ecosystems where they can 're-learn' social, foraging, and predator-defence behaviours (Goderie et.al., 2013) The Tauros de-domestication programme, led by the non-profit Grazelands Rewilding (previously Stichting Taurus), aspires to a future where an Auroch-like species roams freely as part of megafauna-led ecosystem recovery areas in sparsely populated regions of Europe. However, the programme also recognises the practical need for rewilding across a variety of landscapes, acknowledging that some Tauros populations may need to retain domestic traits, such as docility, in areas used for human recreation and other activities.

Rewilding's pragmatic focus on restoring functional traits often conflicts with the conservation purist view that genetic fidelity within species or breeds is necessary for optimal post-release performance. Proponents of the purist approach argue that genetic proximity to the original species reduces the risk of unexpected functional effects and negative ecosystem outcomes (Thomas, 2013; Seddon et al., 2014; Shapiro, 2015; IUCN, 2016). This tension is particularly evident in the recovery of North American bison, which, despite some introgression of cattle genes (Stroupe et. al., 2022) never became fully extinct allowing conservationists to aim for rebuilding pure-bred herds. However, rewilders seeking to rapidly scale ecosystem recovery are comfortable using bison herds that retain cattle genes from historic cross-breeding, as they are more plentiful, easier to source, and appear to express the same phenotype and functional effects (Preston, 2024). This purist-pragmatist tension is less pronounced in European bovid de-extinction, as it begins from a fully domesticated starting point.

Rewilding tends toward a more hands-off and 'fluid' approach, placing trust in the capacity of large herbivores with restored functional traits to naturally reintegrate into ecosystems that have been released from intensive anthropogenic pressures. Rather than striving for genetic replication, the focus of large herbivore de-extinction is on allowing 'wilded' taxa to co-create ecosystems that, while resembling the past, have the capacity to evolve in novel and dynamic ways.

## De-extinction, natural processes and nature-based solutions

Rewilding's focus on restoring functional traits in large herbivores stems from its origins and position in Europe as a nature-based solution (NbS) to complex environmental and societal challenges (Jepson et al., 2021). Since the 2009 Copenhagen Climate Conference (COP15), where the IUCN highlighted the role of ecosystems in climate adaptation and mitigation (Cohen-Shacham et al., 2016), the concept of NbS has been integrated into core climate, biodiversity and land-use policies, as well as corporate sustainability and sustainable investment strategies (Seddon, et al., 2020; Davies et al, 2021; Mayor et al., 2021).

A growing body of rewilding science investigates the relationship between the functional effects of megafauna, natural (ecosystem) processes, and the ecosystem services and societal benefits derived from the recovery of ecosystem integrity. Despite their frequent mention in the scientific literature, natural processes remain poorly theorised and classified. These processes involve the movement or transfer of energy, materials, or organisms within ecosystems, driving functions and dynamics that influence ecosystem condition, recovery, resilience, and long-term evolution. While natural processes encompass a wide range of interactions—such as photosynthesis, predation, and pollination – a land-mark paper by Perino et al. (2019) posited that improvements in three higher-order natural processes, namely dispersal, random disturbance and trophic cascades would support and accelerate the recovery of complex ecosystems. The authors emphasised the central role of large herbivores in the recovery of these processes.

Building on this, scientists and rewilding practitioners have demonstrated how restored are large herbivore populations drive the recovery of key ecosystem processes such as nutrient cycling, seed and nutrient dispersal, permafrost cooling (albedo effect), and soil and hydrological processes (see, e.g., Cromsigt et al., 2016; Macias-Fauria et al., 2021; Kristensen et al., 2022). The interactions between their functional effects, embodied energy and resources, and natural processes create emergent properties leading to complexity (e.g. via niche creation) and structures (e.g. scavenger ecologies, Rewilding Europe/Ark Nature, 2017).

Rewilding draws attention to the downgrading legacy of large herbivore extinctions on ecosystem integrity and the opportunity to reverse this decline through de-domestication pathways. The state of ecosystem integrity directly correlates with the quality and quantity of ecosystem services, which provide societal benefits such as hazard reduction, pollution control, and human well-being. Many of these benefits result from joint production processes involving both ecosystem and human inputs, mediated by new institutional frameworks such as markets for environmental credits. De-domesticated cattle and horses, at various stages on the 'wilding' pathway, are helping create ecosystem assets that sequester carbon (Burrell et al., 2024), reduce the risk of extreme flood and wildfire events (Jepson et al., 2018; Johnson et al., 2018), and contribute to rural regeneration through ecotourism (Hall, 2019).

In the context of rewilding and NbS, the purpose of large herbivores de-extinction via de-domestication pathways is to revive ecosystem processes and functions, thereby enabling ecosystems to recover their integrity and capacity to provide ecosystem services and benefits for society This shifts the moral imperative of de-extinction from redemption for past human-induced extinctions to creating opportunities for co-developing ecosystem assets together with non-human life forms. Rewilding offers a compelling policy rational for de-extinction and addresses criticisms that it is hubristic (Odenbaugh, 2023), risks diverting resources from urgent conservation priorities (Bennett et al., 2017), or undermines conservation efforts by providing a perceived 'offset' for environmental damage (Sandler, 2013).

## The need for an enabling policy environment for large-herbivore de-extinction

This functionalist approach to ecosystem restoration challenges current regulatory frameworks and requires innovative policies that support ecosystem-based approaches. The de-domestication of large herbivores as ecosystem engineers has no precedent in policy. In Europe, large large-bodied animals are assigned policy identities that regulate how they live and how humans interact with them. Large herbivore populations may be classed as wildlife, game and/or livestock. However, due to their ancient history of domestication and extinction, cattle and horses are culturally extinct as both wildlife and game animals. As a result, all breeds and populations are classed as livestock and their de-domestication must proceed within a complex and comprehensive system of European livestock regulations developed for intensive livestock production systems. These regulations include stringent controls on animal husbandry, movement, and practices such as individual identification, disease testing, and carcass disposal.

This regulatory regime is clearly in tension with the NbS purpose of rewilding and large-herbivore de-extinction, and its implementation is more stringent for bovines whose products enter the food chain. The de-domestication process seeks to restore animals' ability to express natural behaviours, roam freely, and adapt to their environments. However, current livestock regulations—such as tagging and health checks—create significant barriers. These regulations necessitate specialised infrastructure and handling methods, which conflict with the goal of re-establishing self-sustaining wild herds.

Affording de-domesticated cattle (such as Tauros) wild status is unlikely in the foreseeable future, partly due to institutional inertia and concerns about de-extinction's validity in biodiversity conservation (Genovesi and Simberloff, 2020). Wild status may also be undesirable for pragmatic reasons. This is because active management is needed during the "wilding" phase of de-domestication when founder populations are small and naive. This involves continued selective breeding, mitigating the Allee effect (the negative relationship between population growth rate and small population size), and providing supplementary feeding and predator protection while animals relearn herd behaviours and rebuild their ecological "life spheres."

Additionally, to fulfil the role of ecosystem engineers in areas where ecosystem services are expected, society will likely demand that individuals or organisations be accountable for the actions and care of wilded large herbivores. This will include their contribution to ecosystem recovery and reducing perceived suffering (e.g., from winter starvation) to avoid controversy (Theunissen, 2019). Such expectations of ownership are currently inconsistent with the notion of wild status in Europe. However, they may be less pronounced for "wilded" horses due to the existence of feral and semi-wild populations in different regions of Europe.

A potential solution to circumvent restrictive livestock regulations is to advocate for a new "kept wild" category of livestock within agricultural regulatory frameworks, supported by policies designed to support and enable their role as ecosystem engineering assets. To frame this policy innovation, I propose adopting the label "eco-herd" to refer to cattle and other large-bodied herbivores bred

and managed specifically for their functional role in ecosystems. I use the term "herd" because it is the social group, not the individual (or species), that generates system-level effects. An eco-herd could be defined in the policy as a "social group with the autonomy to fulfil functional roles in ecosystems in keeping with their evolutionary traits."

Focusing on herds, rather than species, would allow de-domesticated large herbivores to be managed based on their ecological contributions rather than their taxonomy. This approach aligns with area-based strategies in agricultural and land-use policy: specific regions could be designated as ecosystem restoration areas where tailored regulations apply to eco-herds. Criteria for classifying eco-herds could be based on functional effect schemas (as outlined earlier) and herd population and area size. This policy innovation would create an enabling policy environment for large herbivore de-extinction following de-domestication pathways.

## Concluding remarks

Rewilding practitioners are actively de-domesticating cattle and horses to restore their functional roles within ecosystems and to develop nature-based solutions for climate adaptation and rural renewal. European rewilding is based on the ethos that there is no way back, only forward, and the meaning of the 're-'prefix is 'again' rather than 'back.' In the context of rewilding, the purpose of large herbivore back-breeding is not to recreate an exact replica of an extinct genome and phenotype, but to restore free-living social herds capable of co-creating rich, self-sustaining, and complex ecosystems. This requires back-breeding toward extinct phenotypes to remove genes that express traits restricting a taxa's ability to thrive and evolve in recovering ecotypes. In cattle, these include genes for large rump body mass and udders, smooth hides and small or absent horns. These morphological traits increased risk of injury, predation and starvation and reduce a breeds' functional effects.

The term 'de-extinction' sits uncomfortably with rewilding because it implies the undoing or reversing of a state of loss, which is not scientifically or practically achievable. As many have noted 'de-extinction' increasingly refers to creating proxies or functional equivalents of extinct species. Rewilding aligns with Novak's (2018) definition of de-extinction, which emphasises the ecological replacement of extinct species through the purposeful alteration of phenotypes using breeding techniques with the goal of restoring '*vital ecological functions that sustain dynamic ecosystems and increase biodiversity and bio-abundance*'. (p. 5).'

Novak (ibid.) uses the term 'proxy,' which implies a substitute. However, I recommend the term 'analogue' because it better acknowledges the individuality and agency of de-domesticated large herbivores. Rewilding projects focus on de-domesticating bovids and equids as active players in the recovery of ecosystems that will be novel yet resembling past natural baselines. These taxa are not mere substitutes, they are active players in shaping dynamic and novel ecosystems. The adoption of vernacular names such as Tauros or Auroch 2.0 reflects the rewilding philosophy that species and ecosystems, like societies, are always in states of 'becoming'.

In my view, the grammatical tension between the prefix 'de-' (implying reversal) and the finality of the verb 'extinction' (denoting a definitive state) becomes less significant when the scope of de-extinction is extended to include the revival of a taxon's functional effects and *Umwelt* (Uexküll, 1909) – the sensory bubble or 'life-sphere' that shapes and constrains an animal's life, role, and future in ecosystems. By reversing the decline and narrowing of large herbivore *Umwelten* caused by domestication, rewilding restores the autonomy of these taxa and with this the autonomy of ecosystem processes and functions. This restoration enhances the capacity of an ecosystem to recover and adjust to climate and wider environmental changes that are too complex for human management. Expanding the scope of extinction discourse in this way is consistent with the broader shift in conservation science and practice from a compositionalist approach – focusing on entities and components – to a functionalist approach that emphasises system interactions and the reconnection of nature and society.

In their thought-provoking critique of de-extinction, Banks and Houchuli (2017) observed that every cause needs its icons, rallying points and symbols. They argued that de-extinction risks undermining the value of extinct species as martyrs for the conservation cause. This may be true, but in an era of eco-anxiety, the conservation cause also needs heroes and symbols of hope. Rewilding offers a fresh and empowering environmental narrative (Jepson, 2019) where a cast of megafauna – whether self-recovered, reintroduced, or de-domesticated – serve as 'heroic' and charismatic characters in stories of recovery, renewal and transformation. These animal characters and their stories of de-extinction and recovery can help restore collective confidence in the future.

**Open peer review.** To view the open peer review materials for this article, please visit http://doi.org/10.1017/ext.2024.27.

**Acknowledgements.** I would like to thank Ronald Goderie and Eric Schllekens of Grazelands Rewilding for the many conversations and opportunities to observe and discuss different aspects of their Tauros programme. I am also grateful to Frans Schepers, Wouter Helmer, Deli Saavedra, and Pedro Prata for sharing their extensive knowledge of horse and bovid rewilding and for providing opportunities to visit rewilding areas. Lastly, I would like to thank Rhys T. Lemoine and an anonymous reviewer for their valuable comments, which improved this article.

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
