## [Reviewer Report]

Comments on “De-Extinction Beyond Species: 3 Restoring Ecosystem 4 Functionality through Mega- 5 herbivore Rewilding”

10 - Generally the definition of “megaherbivore” is restricted to species with mean mass ≥1,000 kg. Just megafauna or large herbivores/grazers would be more appropriate. De-extinction is also maybe an exaggeration when talking about phenotypic selection (or in the case of horses, really just natural selection) in extant species, but I understand that both have already been used throughout in the paper. De-domestication, used elsewhere in the paper, or breeding-back might be more appropriate.

13 - Wording is a little awkward, maybe remove “assets for” and replace “to” with “for” so that it just reads “acting as nature-based solutions for complex…”

14 - “policy rationale” is a little awkward, maybe “rationale in policy”?

21 - The inclusion of scientific names, e.g. (Bos taurus/primigenius, Equus caballus/ferus) would be appropriate here I think. Obviously it’s complicated since the convention is fuzzy on treating domesticated species as separate species or subspecies (I would heavily favour the latter) but I think it is necessary to include the taxonomy.

21 - Would also use “for” instead of “of”.

27 - Definitely a comma or an “and” missing here somewhere

85 - alter or altering?

96 - function of?

99 - comma, not period

99 - I would refrain from using the word “tarpan”, as that refers to a specific population that we now know to have been at least partially domestic https://doi.org/10.1038/s41586-021-04018-9

117 - The Tauros program is only one such program - there is also the Auerrind project, as well as the original Heck projects and their current incarnation the Taurus (emphasis on the u instead of an o) project

121 -I would be careful not to lean on the genetic component of the project, as it really has very little to do with what they are actively practicing, which is really just selective breeding

135 - Worth noting that “pure” bison without cattle introgression may not exist https://doi.org/10.1038/s41598-022-09828-z

173 - seed and nutrient dispersal?

206 - policy identities? Political identities?

General notes

There are a fair number of mistakes in the language which impede readability in some parts, particularly with sentences that feature commas, and the paper would benefit from some editing in that regard.

A lot of focus is placed on cattle and horses, understandable since these have wild equivalents that no longer exist (though debatable with horses since Przewalskis exist and are the same species, albeit not the subspecies from which horses were domesticated, and are also used in European rewilding projects). However, I wonder if much of this doesn’t also apply to other relevant species like pigs (used in several projects in place of extant but locally absent wild boar) or water buffalo (used in several projects, often not explicitly as an analogue but conceivably a substitute for extinct Bubalus murrensis or near-east populations of the Asian species from which the domestic variety descends).

Overall I think this is a timely paper with a good message that would benefit from clearer definitions and intentions. I think a lot of attention should be paid to the language in the paper to make sure not only that it has fewer errors but that it more clearly conveys the ideas being presented. Consequently, I would recommend some revision in these aspects.

Looking forward to seeing the final version and best of luck

Rhys T. Lemoine, PhD, University of Gothenburg

---

## [Reviewer Report]

Apart from minor textual things (I will send an pdf with remarks to author and to Cambridge Prisms) I fully agree with the chosen pragmatical approach to consider rewilded cattle as ‘ecoherds’. And I fully agree with the line that the ecological functioning (wild behaviour is central) of replacements for lost wild species of current-day domestic animals is key. Not so much whether or not the genes match 100%. Of course the chances for survival in wild landscapes will be larger if there is a match with the wild-type animals, since those features apparently helped to become the best fit (in an evolutionary perspective). So the closer you look and behave as an auroch, the better your chances for survival in wild landscapes. The discussion on genetic purity - as in the American bison is in my opinion contraproductive and might lead to an ever smaller genetic base. The ecological functioning - in a modern context - should be central. And that’s the approach of this article as well.

---

## [Editor Report]

Dear Dr. Jepson,

Thank you for submitting your manuscript to Cambridge Prisms Extinction for consideration. As you’ll see the reviewers were generally positive and had only minor suggestions for revisions. Please note that reviewer 2 sent a pdf with comments. If it isn’t successfully sent with this email, please let me know and I’ll ask the journal office to send it to you directly.

I look forward to seeing your revised manuscript.

Best wishes,

Kate Lyons

Handling Editor

---

## [Editor Report]

Dear Dr. Jepson,

Thank you for submitting your revised manuscript to Cambridge Prisms-Extinction. I appreciate the care you took in addressing the reviews. I am recommending that we accept your manuscript for publication as part of the special issue on Rewilding. 

Best wishes,

Kate Lyons